# HouseLayout3D: A Benchmark and Training-Free Baseline for 3D Layout Estimation in the Wild

**Valentin Bieri**
ETH Zurich

Marie-Julie Rakotosaona
Google Research

Keisuke Tateno
Google Research

Francis Engelmann
Stanford University

Leonidas Guibas
Stanford University

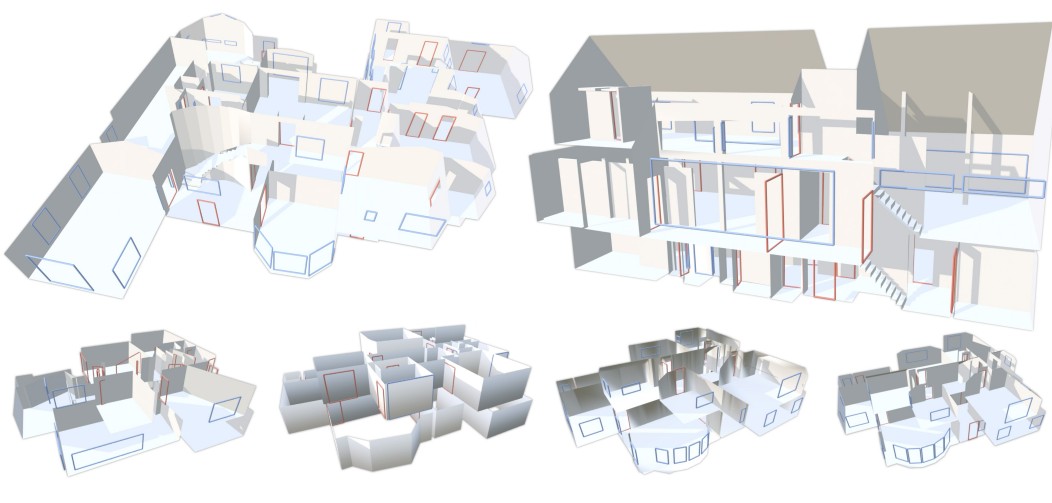

*RoomFormer* [1]    *SceneScript* [2]    *MultiFloor3D* (Ours)    *Ground truth*

Figure 1: **Top:** We introduce the HOUSELAYOUT3D dataset, a benchmark for 3D house layout estimation which is more diverse than existing datasets and includes large-scale multi-floor buildings and annotations for doors, windows and staircases. **Bottom:** We propose MultiFloor3D, a training-free method for 3D layout estimation that improves over existing methods on our and existing datasets.

## Abstract

Current 3D layout estimation models are predominantly trained on synthetic datasets biased toward simplistic, single-floor scenes. This prevents them from generalizing to complex, multi-floor buildings, often forcing a per-floor processing approach that sacrifices global context. Few works have attempted to holistically address multi-floor layouts. In this work, we introduce HOUSELAYOUT3D, a real-world benchmark dataset, which highlights the limitations of existing research when handling expansive, architecturally complex spaces. Additionally, we propose MultiFloor3D, a baseline method leveraging recent advances in 3D reconstruction and 2D segmentation. Our approach significantly outperforms state-of-the-art methods on both our new and existing datasets. Remarkably, it does not require any layout-specific training. The HOUSELAYOUT3D dataset and evaluation scripts are available on the project page: https://houselayout3d.github.io

## 1 Introduction

Estimating the layout of 3D scenes is essential for several computer vision and robotics applications [1, 2, 3, 4]. The objective of 3D layout estimation is to convert a 3D space into a compact, vectorized representation. Specifically, we seek to abstract a reconstructed 3D mesh into a set of closed polygons that define structural elements such as walls, floors, and ceilings, along with doors, windows, and

39th Conference on Neural Information Processing Systems (NeurIPS 2025) Track on Datasets and Benchmarks.

staircases, while disregarding occluding objects like furniture, which commonly appear in real-world environments.

Recent state-of-the-art models for layout prediction [1, 2, 5] are feed-forward deep-learning models trained on large-scale synthetic datasets [6, 2] and demonstrate impressive results even on real-world scenes. A key aspect of these models is that they are trained on synthetic data, which primarily consists of single rooms or small apartments. This is largely because such smaller scenes are easier to synthesize—they can be automatically generated at scale [3] or designed by professionals [6]. As a result, models trained on this data face significant limitations, struggling to generalize to large-scale buildings with substantially more rooms than a typical apartment and being entirely incapable of handling multi-level or multi-floor buildings. While it is possible to first divide large-scale buildings into individual floors and rooms and then process them separately, this approach discards valuable global context that can aid in local reasoning. For instance, detecting structural elements like staircases requires cross-floor reasoning, which is lost when floors are processed in isolation. Additionally, this method necessitates recombining individual room predictions to support building-level tasks such as path planning between rooms on different floors.

To advance research in 3D layout prediction for large-scale, multi-floor buildings, we introduce HOUSELAYOUT3D, a challenging benchmark dataset. Built upon real-world building scans from the Matterport3D [7] dataset, it captures expansive, architecturally complex spaces with up to five floors and forty rooms per floor, encompassing diverse room types, including partially open spaces that pose challenges for existing room-based approaches. We manually annotate all structural elements, including walls, floors, ceilings, staircases, as well as windows and doors, specifying the direction in which each door opens.

Inspired by the success of recent reconstruction and segmentation models, we propose a training-free approach called MultiFloor3D. Our goal is to demonstrate that by leveraging recent advances in 3D scene reconstruction, Gaussian Splatting models, and an innovative layout fitting technique, we can develop a simple yet effective method that outperforms existing approaches on the more challenging task of 3D layout estimation in multi-floor buildings. Our experiments on HOUSELAYOUT3D clearly highlight the limitations of current state-of-the-art methods in handling complex multi-floor buildings. In contrast, our approach generates more accurate and reasonable layouts, particularly for challenging multi-floor structures. We hope that these findings together with the benchmark dataset will inspire new research directions in multi-floor, large-scale 3D layout estimation.

In summary, our contributions are:

- We introduce HOUSELAYOUT3D, the first benchmark dataset for 3D layout estimation in large-scale, multi-floor buildings.
- We propose MultiFloor3D, a training-free baseline method that leverages recent reconstruction and segmentation techniques, achieving improved performance over current deep-learning models.
- Our extensive experiments clearly reveal the limitations of existing layout estimation methods, which we hope will drive further research in this direction.

## 2  Related Work

**Manhattan Scene Layout**   Initial works on layout estimation impose Manhattan world assumptions on the output to then solve a constrained optimization problem based on detected walls (Scan2Bim [8]) or corners (DuLaNet [9], LayoutNet [10], FloorNet [11]). Notably, Ochmann et al. [12] allow angled walls by subdividing the 3D space into cells, ultimately determining the indoor space with an integer linear program.

**2D Scene Layout**   Another line of work solves the problem from Birds-eye View (BEV): [13] uses shortest-path algorithms around the free space. Floor-SP [14] extends the concept with a room segmentation network. HovSG [15] combines 2D BEV point density maps with 2D object detection to build a scene graph of floors, rooms, and objects without predicting their geometry. This line of work is limited by its 2D predictions.

**3D Scene Layout.**   Recent advances were made by end-to-end deep learning methods: SceneCAD [3] uses a graph neural network to infer a 3D layout and object bounding boxes. Room-Former [1] trains a transformer to estimate a 2D floorplan enriched with semantics. SceneScript [2] proposes a *structured scene language* to predict 3D layout walls, windows, doors, and object bounding boxes from sparse point clouds. Importantly, available training data for end-to-end trainable methods is dominated by individual room scenes [3] or simple individual floors [6][2][16]. Moreover, the

| Dataset | Real-world | Multi-room | Multi-floor | Full Scenes | Windows, Doors | Objects | Depth | 3D Layouts |
|---|---|---|---|---|---|---|---|---|
| SceneCAD [3] | ✓ | (✓) | ✗ | ✓ | ✓ | ✓ | ✓ | ✓ |
| ASE [2] | ✗ | ✓ | ✗ | ✓ | ✓ | ✓ | ✓ | ✓ |
| Stru3D [6] | ✗ | ✓ | (✓) | ✓ | ✓ | ✓ | ✓ | ✓ |
| Zillow Indoor [16] | ✓ | ✓ | (✓) | ✓ | ✗ | ✗ | ✗ | ✗ |
| MP3D-Layout [18] | ✓ | ✗ | ✗ | ✗ | ✓ | ✓ | ✓ | ✓ |
| Zou et al [17] | ✓ | ✗ | ✗ | ✗ | ✗ | ✗ | ✓ | ✓ |
| CADEstate [4] | ✓ | ✓ | ✗ | ✗ | (✓) | ✗ | ✗ | ✓ |
| FloorNet [11] | ✓ | ✓ | ✓ | ✓ | (✓) | ✗ | ✗ | ✗ |
| HOUSELAYOUT3D (Ours) | ✓ | ✓ | ✓ | ✓ | ✓ | ✓ | ✓ | ✓ |

Table 1: **Dataset Comparisons** of existing dataset benchmarks for evaluating 3D layouts estimation.

buildings are often unfurnished [16], synthetic [6][2], or limited to Manhattan layouts [17]. Another line of datasets annotates extracts of larger scenes in single 2D images or videos [4][18]. We find that the limited availability of training data prevents end-to-end methods from generalizing beyond simple layouts.

# 3 The HOUSELAYOUT3D Dataset

We introduce a new dataset of hand-annotated CAD layouts derived from the Matterport3D [7] (MP3D) dataset (see Fig.2). Unlike previous works[3, 2], this is the first real-world benchmark dataset to provide CAD annotations for large-scale, multi-floor houses, encompassing numerous rooms, staircases, windows, and doors. Each structural element is annotated as a polygon in 3D space. Since our dataset is annotated on 3D meshes from MP3D [7], it inherits their per-vertex room ids and object instances.

**Dataset Statistics.** The dataset includes 16 buildings, 33 distinct levels, and 317 rooms, captured across more than 26,000 RGB-D frames. Its scale is comparable to the validation split of ScanNet [19]. In total, we annotated 292 doors, 379 windows, and 34 staircases. The lower number of doors compared to rooms is due to many spaces, such as hallways and dining areas, being

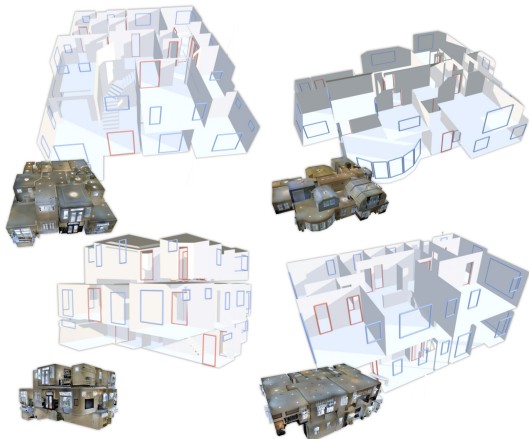

Figure 2: **Examples of our HOUSELAYOUT3D.** Our dataset includes multi-floor houses with annotations for walls, floors, ceilings and stairs, as well as windows *(blue)* and doors *(red)*. We also show the corresponding 3D meshes from MP3D [7].

connected by open passages or staircases rather than actual doors. Each building comprises between 1 and 5 levels and contains between 4 and 40 rooms. The annotation time varies depending on the building's size and the number of rooms, typically ranging from 4 to 10 hours per building. All annotations undergo visual verification by separate expert annotators. Table 1 compares properties across different datasets.

**Annotation Tool and Labeling Details.** To annotate the 3D scans, we use a free academic license of Scasa's PinPoint [20], a specialized software for building modeling from point clouds. It enables precise 3D geometry extraction even in occluded or incomplete areas through intuitive tools that automatically snap to edges and corners, streamlining the annotation process. In the 3D scans, doors are typically open, so we annotate both the current open position and the expected closed position, along with the opening direction. For doors that appear closed in the scans, we infer the opening direction by from the door hinge locations in the RGB images. For window annotations, we utilize the existing window object annotations from MP3D [7], projecting them onto the nearest annotated wall plane and fitting axis-aligned rectangles.

# 4 Method

Given N input RGB images of a scene, our goal is to produce a simple 3D layout consisting of polygons. Each polygon is assigned a label from a finite set of classes: walls, floors, ceilings, stairs, doors, and windows. The layout is organized into a scene graph with rooms as nodes and doors/stairs as edges, and each layout polygon is assigned to a room or an edge of the scene graph.

Figure 3 provides an overview of our approach, which consists of four stages. First, we compute a 3D mesh of the scene. In the second step, we extract the scene's main structural elements (floors,

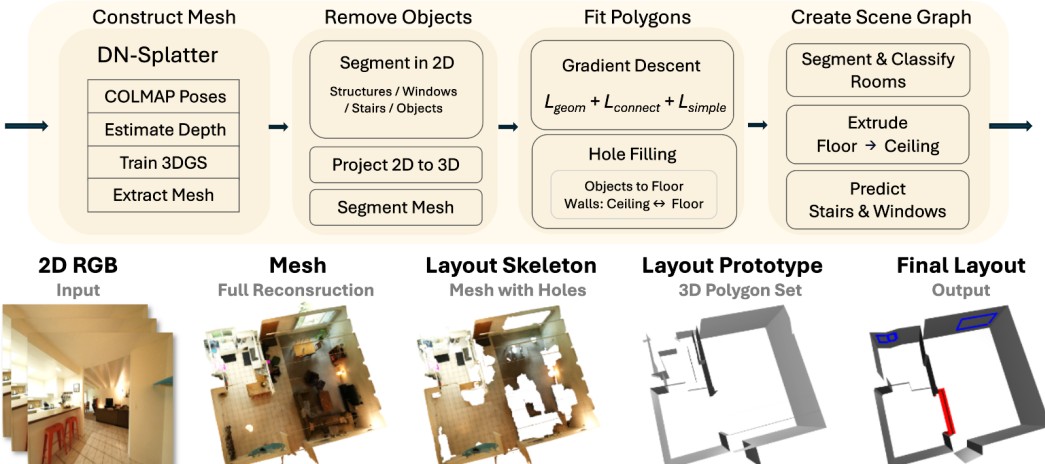

Figure 3: **Illustration o the MultiFloor3D model for 3D layout estimation.**

walls, ceilings) to form a *skeleton* of the layout. In the third step, we use geometric and semantic information to fit a layout *prototype* to the *skeleton*. Lastly, we parse the *prototype* into a scene graph, from which we extract the final layout.

## 4.1 Generating a Mesh from RGB Images

Given a set of unposed 2D images, we follow DN-Splatter [21] to obtain a triangle mesh and 3D depth maps for every frame. DN-Splatter uses COLMAP [22] camera poses and a 2D depth model to train 3D Gaussian Splatting [23] (3DGS) reconstruction. DN-splatter then produces a Poisson Reconstruction [24] by sampling from the 3DGS rendered depth. In this work, we use the depth model Metric3d [25].

## 4.2 Extracting a Layout Skeleton from the Mesh

Once a mesh is generated, our next step is to use a pre-trained 2D segmentation model to extract a minimal, reliable geometry that serves as a basis ('*skeleton*') for the layout. This skeleton should consist exclusively of geometry that we want to include in the final layout. To distinguish such geometry, we define four semantic classes that we treat differently:

- **Structural Components** (*i.e.* walls, ceilings, and floors, but also large furniture such as closets): These are the main components of the layout skeleton. The structural components have accurate geometry that we wish to see represented in the final layout.
- **Geometrically inaccurate surfaces** (windows, mirrors): The 3D representation of windows and mirrors is often inaccurate due to noisy depth estimates. We do not wish to keep them in the layout skeleton.
- **Objects** Smaller furniture and objects such as tables or lamps are removed from the layout skeleton. Objects are later on used to complete unobserved areas of the layout.
- **Stairs** Are processed separately from the layout skeleton due to their complexity.

To construct the skeleton, we segment the 3D mesh into these classes. We run the segmentation model OneFormer [26] on the input images and map each output class [27] to one of the four semantic classes. To transfer OneFormer's segmentation to the mesh, we back-project $M = 5000$ randomly sampled pixels per image and their respective class to 3D. We collect class votes for each mesh vertex by assigning each back-projected point to the nearest mesh vertex. We further postprocess the obtained segmentation by clustering the mesh vertices into *superpoints*, following [28]'s preprocessing step. Each mesh vertex is then assigned to the most common class within its cluster. The result is a mesh segmented into our semantic classes. We create the *layout skeleton* by selecting only *structural components*, and extract *object* and *stair* meshes.

## 4.3 Fitting a Layout Prototype to the Skeleton

We observe significant artifacts in the layout skeletons, including holes and unobserved regions. For example, areas hidden behind furniture, or areas corresponding to windows are missing. In this stage, we use geometric and semantic information to correct the artifacts and infer a more complete

*layout prototype*. To this end, we run an optimization that aims to improve the completeness of the obtained skeleton: We first initialize a collection of planar 3D polygons $\mathcal{P}$ from the layout skeleton. In particular, each segmented superpoint (see Sec. 4.2) of the skeleton is fitted to one or more planes. We then optimize the *vertex positions* and *plane equations* of the polygons using three main objectives:

- Reconstruct an accurate scene geometry with $L_{\text{geom}}$
- Produce a continuous and connected geometry with $L_{\text{connect}}$
- Produce a mesh with low vertex count with $L_{\text{simple}}$

During the optimization, we constrain the vertices of each polygon to be coplanar. We also allow and encourage polygons to share vertices. The initialization and the implementation of vertex constraints and shared vertices is detailed in the supplementary material.

**Definitions.** Given a polygon $P$ consisting of edges $E$ and a point $p \in \mathbb{R}$ we define the point-to-polygon distance $D_{pp}(P, p)$ as the minimal distance between $p$ and any point on the surface of $P$. For $e \in E$ we define the point-to-edge distance $D_{pe}(p, e)$ as the minimal distance between $p$ and any point on the line segment representing $e$.

**Losses.** We fit the polygon set $\mathcal{P}$ using gradient descent and three losses. The first loss $L_{\text{geom}}$ encourages the polygons to reconstruct the original geometry and respect the *observed empty space*:

$$L_{\text{geom}} = L_{\text{prox}} + L_{\text{empty}} \tag{1}$$

$L_{\text{prox}}$ penalizes the distance of each vertex $v \in V_{\text{skeleton}}$ of the Layout Skeleton to the closest polygon surface:

$$L_{\text{prox}} = \sum_{v \in V_{\text{skeleton}}} \min_{P \in \mathcal{P}} D_{pp}(v, P) \tag{2}$$

To prevent occluding the observed empty space (*i.e.* the space we believe to be empty based on the depth maps), we sample a set $L$ of line segments using the input camera poses and computed depth maps. Each line segment extends from the camera pose to the back-projected depth. We then penalize line segment-polygon intersections as follows: If a line segment $l$ intersects a polygon, the nearest polygon edge $e^*$ should be moved closer to the intersection point $p_{inter}$.

$$L_{\text{empty}} = \sum_{l \in L} \sum_{\substack{P \in \mathcal{P} \\ l \cap P \neq \varnothing \\ D_{pe}(p_{inter}, e^*) \leq T_{\text{inter}}}} D_{pe}(p_{inter}, e^*) \tag{3}$$

$$\text{where } p_{inter} = l \cap P \quad \text{and} \quad e^* = \operatorname*{argmin}_{e' \in \text{edges}(P)} D_{pe}(v, e').$$

Note that we ignore intersections with $D_{pe}(p_{inter}, e^*)$ greater than the threshold $T_{\text{inter}}$ to avoid noise from intersections far from the polygon boundary.

The second loss $L_{\text{connect}}$ prevents small gaps and encourages shared boundaries by making polygons attract vertices. Concretely, $L_{\text{connect}}$ penalizes the distance from each polygon vertex to the closest surface of another polygon:

$$L_{\text{connect}} = \sum_{P \in \mathcal{P}} \sum_{v \in \text{vertices}(P)} \min_{P' \in \mathcal{P}, \ P' \neq P} D_{pp}(v, P') \tag{4}$$

As for $L_{\text{empty}}$, we ignore points with $D_{pp}(v, P')$ greater than a threshold.

The third loss encourages simplicity and smooth polygon boundaries. $L_{\text{simple}}$ penalizes the length of all edges that are not shared by at least two polygons. (*i.e.* not all edge vertices are shared). Intuitively, $L_{\text{simple}}$ promotes shared edges (for instance, an edge between two walls) to represent the scene while edges that are not shared are shrunk until they are eliminated.

$$L_{\text{simple}} = \sum_{P \in \mathcal{P}} \sum_{e \in \text{edges}(P)} \mathbf{1}_{[\nexists P' \in \mathcal{P} \setminus \{P\}: \ e \subset P']} \|e\|_2 \tag{5}$$

Our final loss is given by $L = L_{\text{geom}} + L_{\text{connect}} + L_{\text{simple}}$.

**Vertex Merging** $L_{\text{simple}}$ itself does not reduce the number of vertices or polygons in the polygon set. Instead, we periodically manually simplify $P$ by (1) merging vertex pairs with distance below $T_{\text{merge}}$, (2) applying the RDP [29] algorithm with tolerance $T_{\text{merge}}$ to the polygons individually, and (3) merging close polygons with similar normal. (Close in terms of minimal $D_{pp}$ distance among the vertices.) For (3) we additionally verify that the merged polygon does not increase $L_{\text{prox}}$ too strongly. Note that step (1) is the source of shared vertices between polygons.

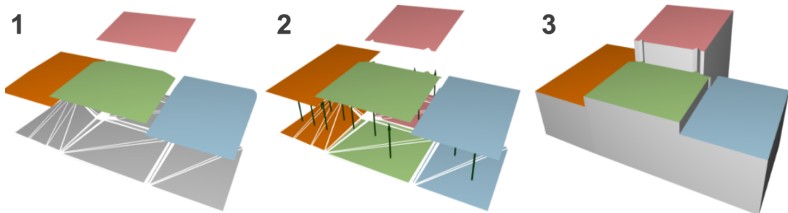

Figure 4: **Our proposed floor extrusion algorithm.** 1) Floor triangulation. 2) Triangles assigned to ceilings using midpoints. 3) Triangles extruded to ceiling planes.

**Closing Holes in the Floor**   We observe that there is a floor under most objects in a room. We exploit this information by projecting objects to the floor, *i.e.*we project each triangle of the *object* mesh extracted in Sec 4.2 to the plane equation of the nearest floor-classified polygon whose centroid lies below the triangle. We recompute the floor polygon from the union of the original floor polygon and the projected triangle surfaces.

**Closing Holes in Walls**   We extend walls to span from ceiling to floor. In particular, we identify polygon edges of wall-classified polygons whose normals face down. Then we count how many line segments in $L$ (representing the observed empty space) intersect the area between each edge and the floor. If the number of intersections per cm$^2$ is below $T_{\text{extend}}$, we extend the edge to the floor. To ceilings and upwards-facing wall edges, we apply the same procedure. We call the output of this stage the *Layout Prototype*

### 4.4   Scene Graphs from a Layout Prototype

In this stage, we convert the prototype (a set of semantically labeled polygons) into the final layout. The final layout is organized as a scene graph, where the nodes (rooms) are connected by edges (doors, stairs). Every node is composed of a single floor, and a set of walls, ceiling, and window polygons. To achieve this, we first create 2D floorplans, which we later extrude into 3D space. The indirection via 2D is motivated by the fact that our 3D layout prototype neither gives us an understanding of indoor/outdoor space nor guarantees a closed or even connected layout.

**Creation of a Scene Graph of 2D Floorplans**   In this step we use the layout prototype and its semantics to (1) identify the different levels (floors) of the building, (2) create a 2D layout (floorplan) of each level, and (3) segment each level into rooms, extracting a per-level 2D scene graph from each floor and (4) detect stairs to connect the individual levels. In the following, we provide an outline of the applied algorithms, which are detailed in the appendix.

- To identify building floors, we use the floor-classified polygons of the layout prototype, merging close levels with similar heights.
- To create a 2D floorplan of each level, we merge each level's floor polygon(s) with suitable ceiling polygons — since ceilings are rarely occluded by objects and thus are more robustly represented in the layout prototype.
- To segment each level into rooms, we apply Hov-SG [15]'s room segmentation algorithm on each 2D floorplan (and the walls of the layout prototype). The segmentation outputs a scene graph with rooms as nodes, and *openings* as edges. We consider an opening edge a *door* if its width is below $1.5\,\text{m}$. Otherwise, we retain its edge but label it as *opening*. Furthermore, each room is associated with a room type (kitchen, office, ...).
- To identify stairs, we cluster connected components of the stair mesh extracted in Sec. 4.2. For each component, we add an edge to the scene graph between the rooms/floors it connects.

**Back to 3D: Room Extrusion**   Sec. D describes how we use the layout prototype to generate a scene graph of rooms. In this section, we propose a simple algorithm inspired by layout annotation tools [20], that extrudes each node's 2D floorplan to the ceiling. For a single room, the extrusion algorithm creates a closed room shell using a 2D floorplan and a set of potential 3D ceiling polygons that at least partially cover the floorplan. Fig. 4 visualizes the extrusion process. Its core idea is to (1) triangulate the 2D floorplan, (2) assign each triangle to a ceiling polygon and (3) extrude each triangle to its ceiling. Specifically, we triangulate the room's 2D floorplan using a 2D Constrained Delaunay Triangulation [30] built from the boundary of the floorplan, the ceiling candidates' edges, and the projections of the pairwise intersection lines of the ceiling candidates' planes. For each triangle center, we cast a ray upward. If the ray hits a ceiling candidate, we assign the triangle to that ceiling's plane. Intuitively, this assignment partitions the floorplan by 'rendering' ceiling polygons on the floor. Triangles that do not hit a ceiling are assigned to the lowest ceiling plane reachable in

| Method | Structures | | Doors | | Windows | | Stairs | | Depth | | |
|---|---|---|---|---|---|---|---|---|---|---|---|
| | F1@0.5 | Avg F1 | F1@0.5 | Avg F1 | F1@0.5 | Avg F1 | F1@0.5 | Avg F1 | $\Delta_5$ | $\Delta_{10}$ | #Vertices |
| RoomFormer [1] (per floor) | $0.24_{\pm0.06}$ | $0.22_{\pm0.06}$ | $0.23_{\pm0.10}$ | $0.20_{\pm0.09}$ | $0.07_{\pm0.06}$ | $0.07_{\pm0.04}$ | – | – | $24.9_{\pm11.5}$ | $32.9_{\pm14.9}$ | 764.9 |
| RoomFormer [1] (per room) | $0.18_{\pm0.14}$ | $0.16_{\pm0.12}$ | $0.18_{\pm0.14}$ | $0.16_{\pm0.12}$ | $0.08_{\pm0.08}$ | $0.09_{\pm0.07}$ | – | – | $37.3_{\pm10.4}$ | $44.8_{\pm10.7}$ | 1134.5 |
| SceneScript [2] (per floor) | $0.28_{\pm0.11}$ | $0.26_{\pm0.08}$ | $0.23_{\pm0.26}$ | $0.20_{\pm0.23}$ | $0.16_{\pm0.18}$ | $0.15_{\pm0.17}$ | – | – | $22.5_{\pm8.6}$ | $33.8_{\pm11.7}$ | **677.1** |
| SceneScript [2] (per room) | $0.23_{\pm0.12}$ | $0.21_{\pm0.11}$ | $0.31_{\pm0.26}$ | $0.28_{\pm0.23}$ | $0.11_{\pm0.11}$ | $0.10_{\pm0.09}$ | – | – | $23.5_{\pm7.2}$ | $32.9_{\pm6.7}$ | 1333.6 |
| MultiFloor3D (Ours) | $\mathbf{0.40_{\pm0.10}}$ | $\mathbf{0.38_{\pm0.10}}$ | $\mathbf{0.55_{\pm0.16}}$ | $\mathbf{0.44_{\pm0.15}}$ | $\mathbf{0.43_{\pm0.29}}$ | $\mathbf{0.38_{\pm0.22}}$ | $\mathbf{0.42_{\pm0.48}}$ | $\mathbf{0.41_{\pm0.44}}$ | $\mathbf{61.1_{\pm9.2}}$ | $\mathbf{76.3_{\pm7.9}}$ | 1957.0 |

Table 2: **Scores on HOUSELAYOUT3D.** Performance comparison with state-of-the-art layout estimation methods in terms of average and standard deviation across scenes. Structures include wall, floor and ceilings. MultiFloor3D is the only method predicting stairs.

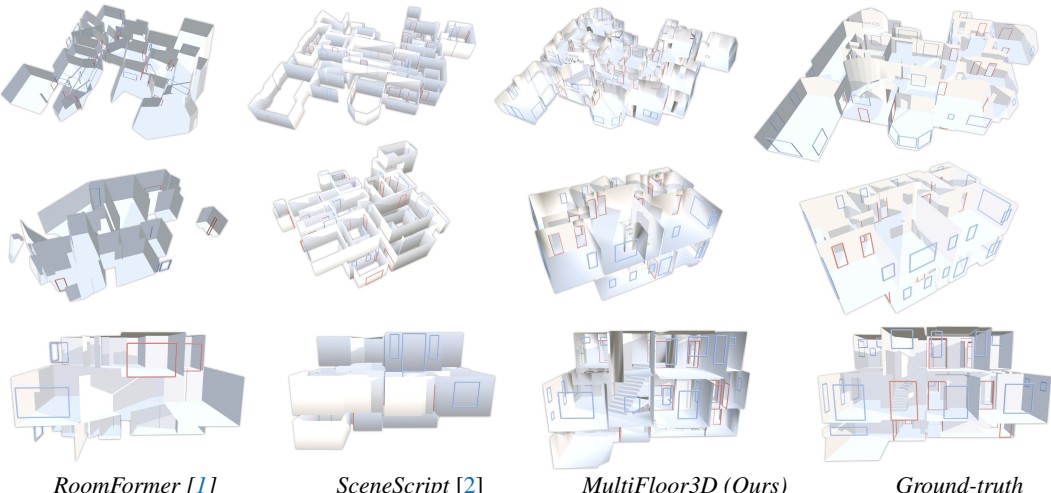

*RoomFormer [1]*     *SceneScript [2]*     *MultiFloor3D (Ours)*     *Ground-truth*

Figure 5: **Qualitative Results on HOUSELAYOUT3D.** We present layout estimation samples from our model alongside state-of-the-art methods. To enhance visualization, we apply back-face culling to the layout meshes, allowing a clear view inside the buildings. Since SceneScript represents walls as boxes, back-face culling is ineffective; instead, we remove the added floors and ceilings for better visibility.

the graph of unassigned triangles. (Lowest in terms of the triangle midpoint's projected z-coordinate on the target plane.) Lastly, we extrude each floor triangle to its assigned ceiling plane. That is, we produce ceiling and floor triangles on the ceiling and floor planes respectively, and add axis-aligned wall rectangles for triangle edges coinciding with a wall in the 2D floorplan. To ensure a closed room shell, we further add vertical rectangles along potential discontinuous edges in the extruded ceiling surface. To limit complexity, we only consider the 30 largest ceilings per room. Details on how we add doors and stairs after extruding are provided in the appendix.

**Window Detection** To detect windows, we back-project the 2D window segmentation of the input images obtained in Sec. 4.2 onto layout walls and cluster the result. Concretely, we create rays for window-classified pixels and intersect them with the walls of our 3D layout. We then filter outliers [31], split the points by wall instance, and run DBSCAN [32] for each wall to identify window clusters. To every cluster with at least $k = 10$ vertices, we fit an axis-aligned bounding rectangle. Finally, we predict a window for every rectangle with height and width greater than 30cm.

## 5 Experiments

In this section, we first introduce metrics to measure the performance of 3D room layout estimation, and then compare our approach to recent state-of-the-art methods on the proposed MultiFloor3D dataset (Sec. 5.1) as well as ScanNet++ [33] (Sec. 5.2). We then provide analysis experiments to understand the importance of the individual pipeline components (Sec. 5.3), and conclude with qualitative results and potential applications (Sec. 5.4).

**Methods in Comparison.** We compare our approach with two recent methods for scene layout estimation, namely RoomFormer [1] and the recent SceneScript [2]. Training these baselines on our multi-floor dataset is non-trivial – RoomFormer is designed for 2D floorplan prediction, while SceneScript is limited to 4-corner primitives. Instead, we evaluate them using their publicly available model weights on the full HOUSELAYOUT3D dataset. Both baselines are trained on large-scale synthetic datasets (∼100k samples), whereas our approach is training-free. Similar to [2], we extrude RoomFormer's 2D predictions to 3D. Finally, as neither baseline explicitly predicts ceilings or floors, we append ceiling and floor polygons to each predicted room to ensure a fair depth evaluation.

| Method | #Vertices | $\Delta_5$ | $\Delta_{10}$ |
|---|---|---|---|
| DN-Splatter Mesh [21] | 354k | 84.1 | 92.6 |
| RoomFormer [1] | **32.5** | 36.8 | 48.9 |
| SceneScript [2] | 41.2 | 55.1 | 68.5 |
| MultiFloor3D (Ours) | 83.1 | **67.8** | **84.7** |

Table 3: **Scores on ScanNet++ [33].** Metrics evaluate depth accuracy as an approximation of layout estimation error. Scores are averaged over validation scenes.

| Method | Avg F1 | #Vertices | Sem. |
|---|---|---|---|
| Input Mesh + QSlim [35] | 0.109 | 2000.0 | ✗ |
| Layout Skeleton + QSlim [35] | 0.223 | 2000.0 | ✗ |
| Layout Prototype | 0.373 | 2553.0 | ✗ |
| MultiFloor3D (Ours) | **0.381** | **1957.1** | ✓ |
| (w/o prototype fitting) | 0.214 | 2269.8 | ✓ |
| (w/o room segmentation) | 0.359 | 2442.2 | (✓) |

Table 4: **Ablation Study on HOUSELAYOUT3D.**

**Layout Metrics.** To assess the accuracy of the estimated layouts, we adopt the F1 score based on the *entity distance* $d_E$, following SceneScript [2]. This metric measures the alignment between ground truth entities $E$ and predicted entities $E'$. For rectangular entities (*e.g.* doors and windows), $d_E$ is computed as the maximum distance between corresponding corners of two rectangles of the same class: $d_E(E, E') = \max\left\{\|c_i - c'_{\pi(i)}\| : i = 1, \dots, 4\right\}$ where $\pi(i)$ denotes the optimal corner permutation obtained via Hungarian matching. The F1 score $@\tau$ is then computed by applying a threshold $\tau$ to $d_E$ as in [2].

For non-rectangular entities, we introduce a generalized entity distance $d_H$ which allows comparison between entities with different numbers of corners. We define $d_H$ as the Hausdorff distance between two polygon surfaces (*i.e.* entities) $P$, $P'$ and their vertices $V$, $V'$:

$$d_H(P, P') = \max\left\{\max_{v \in V} D_{pp}(v, P'), \ \max_{v' \in V'} D_{pp}(v', P)\right\} \tag{6}$$

for the point-to-polygon distance $D_{pp}$ defined in Sec 4.3. We then use $d_H$ analogously to $d_E$ to compute the F1 score for walls, floors, and ceilings.

**Depth Metrics.** Following [17], we use input camera poses to render depth maps for the ground truth geometry $D_{GT}$ and predict layouts $D_{\text{pred}}$. When explicit layout annotations are unavailable (*e.g.*, ScanNet++ [33]), depth consistency serves as a proxy for evaluating layout accuracy. Specifically, we compute the percentage of predicted pixel depths that fall within a threshold $T$ cm of the GT depth:

$$\Delta_T = \frac{1}{N} \sum_{i=1}^{N} \mathbf{1}_{[|D_{\text{pred}}(i) - D_{GT}(i)| \leq T]} \tag{7}$$

This $\Delta_T$ metric was introduced [34] and is commonly used in monocular depth estimation [25].

### 5.1 Results on the HOUSELAYOUT3D Dataset

Table 2 shows the main results for the F1-based metrics across semantic classes, and depth metrics on our HOUSELAYOUT3D dataset. For this experiment, we use the camera poses, RGB-D images and mesh of MP3D [7]. As neither baseline is designed for multi-floor layout prediction, we apply them separately per floor (or per room) and then merge the per-floor (or per-room) predictions. We use the ground-truth MP3D level and room segmentation and report scores per-floor and per-room. Our MultiFloor3D does not have access to this privileged information.

MultiFloor3D significantly outperforms state-of-the-art layout estimation methods, despite not using ground-truth floor or room segmentation. While both baselines perform better on individual rooms than full floors, this gap is smaller for SceneScript, which favors compactness (*i.e.*, fewer vertices) at the cost of geometric fidelity.

### 5.2 Results on the Scannet++ Dataset

Table 3 shows additional results on the Scannet++ [33] *DSLR* validation split, consisting of 50 scenes captured with a monocular hand-held camera and COLMAP-generated image poses. As ScanNet++ does not provide ground truth layout annotation, we only report depth metrics as an approximation of the layout error. Since ScanNet++ scenes are populated with objects, we use ground truth semantic annotations to ignore those points during the evaluation, as well as points on windows which are typically not well reconstructed in the ground truth laser scan. As input for all methods, use the mesh provided by the Gaussian Splatting approach DN-Splatter [21] in the first stage of our method (Sec. 4.1). The results indicate that MultiFloor3D outperforms the baselines at the cost of compactness (larger number of vertices).

## 5.3 Analysis Experiments

Table 4 shows the contributions in terms of F1 score of each stage in our approach. Note that the outputs of the first and second stages (*mesh* from Sec. 4.1 and *layout skeleton* from Sec. 4.2) are triangle meshes, which we convert to polygon sets by first applying mesh simplification (QSlim [35]), and then greedily merging adjacent triangles whose normals differ by less than 20°. Performance drops significantly when either layout fitting or room segmentation is removed.

## 5.4 Qualitative Results and Applications

We show qualitative results of our approach in Fig. 5 and compare to RoomFormer [1] and SceneScript [2]. Both baselines methods struggle with large areas consisting of multiple rooms, RoomFormer even more than SceneScript. The baselines ere also inherently limitted to predicting rectangular primitives and cannot represent more complex shapes such as sloped ceilings *(top example)*. In Fig. 6 we visualizes qualitative results when removing loss objectives from the mesh fitting stage introduced in Sec. 4.3.

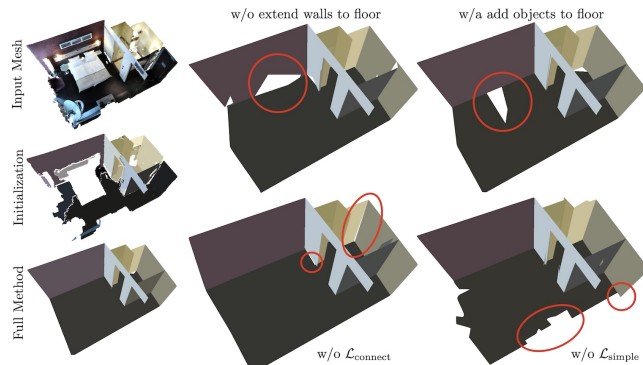

Figure 6: **Effect of Loss Terms.** *Left:* input and output of our approach. *Right:* result when ablating losses and components. Omitting object projection (top center) or wall extension (top right) produces holes in the layout. Without $\mathcal{L}_{\text{simple}}$, the polygon boundaries show dents. Without $\mathcal{L}_{\text{connect}}$, we observe gaps between polygons that otherwise share edges.

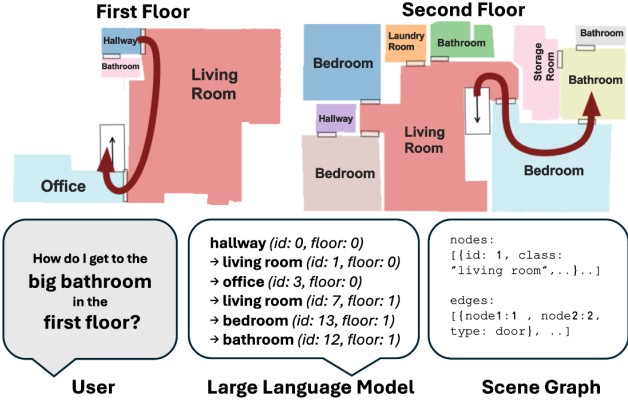

Figure 7: Navigation application based on 3D layouts and LLMs.

**Down-stream Application.** Next, we demonstrate a potential application of full-building 3D layouts. First, we obtain the 3D scene graph where nodes represent rooms, and edges are connections between rooms (doors, stairs, etc.) Then, we feed the scene graph in JSON format to an LLM, together with a user-prompt asking for directions. The LLM responds with turn-by-turn directions on how to reach the desired location. This concept is illustrated in Fig. 7.

## 6 Limitations

MultiFloor3D has a significantly longer runtime than the feed-forward baselines, taking one to two hours per HOUSELAYOUT3D scene on an NVIDIA GeForce RTX 4090, compared to one to two minutes for SceneScript [2] and RoomFormer [1]. Furthermore, MultiFloor3D occasionally struggles to remove outdoor elements perceived through large windows, which can introduce artifacts.

## 7 Conclusion and Discussion

We introduced HOUSELAYOUT3D, the first benchmark dataset for evaluating 3D layout estimation in large-scale, multi-floor buildings. Existing scene layout estimation methods are limited to single-floor buildings, and our experiments reveal their challenges in accurately parsing large-scale floors with multiple rooms—contrasted by our learning-free method, which already outperforms these baselines. Ideally, future research should develop learning-based approaches capable of handling multi-floor, multi-room buildings, rather than relying on heuristics, which, while effective, are significantly slower than feed-forward networks. In summary, we hope that our dataset and evaluation, which highlight the shortcomings of current methods, will drive further advancements in 3D layout estimation beyond single-room and single-level reconstruction.

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

# A Additional Visualizations for the HOUSELAYOUT3D Dataset

Fig. 9 visualizes the scenes in the annotated dataset, while Fig. 8 shows a screenshot of PinPoint [20], the tool used to create the annotations.

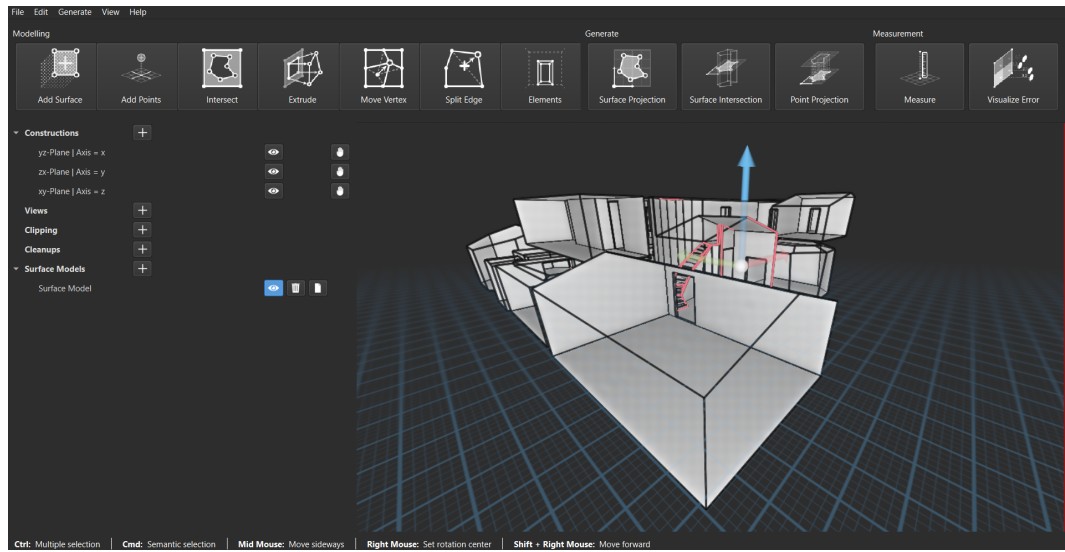

Figure 8: Screenshot of PinPoint [20], the layout annotation tool used to annotate the scenes.

# B Re-Classification of COCO [27] Classes

In Sec. 4.2, we segment the input images using OneFormer [26]. It segments the images into COCO [27] classes, which we re-classify into four semantic classes of interest. Table 5 summarizes this re-classification. For window detection in Sec. 4.4, we make two changes to this classification. Firstly, we do not consider mirrors windows. Secondly, we add the surface classes *window blind* and *curtain* to the window classes. Furthermore, the surface class is maintained in the layout skeleton and the prototype layout (but not in the output).

| Semantic Class | Category | COCO [27] Classes |
|---|---|---|
| Structure | Wall | `wall-brick`, `wall-stone`, `wall-tile`, `wall-wood`, `wall-other-merged` |
| | Ceiling | `ceiling-merged` |
| | Floor | `floor-wood`, `floor-other-merged`, `rug-merged` |
| | Surfaces | `cabinet-merged`, `door-stuff`, `curtain`, `window-blind` |
| Geometrically inaccurate surfaces | Windows | `window-other` |
| | Mirrors | `mirror-stuff` |
| | Outdoor/Noise | `gravel`, `tree-merged`, `sky-other-merged`, `pavement-merged`, `grass-merged`, `dirt-merged` |
| Stairs | Stairs | `stairs` |
| Objects | Object | Rest |

Table 5: Mapping of COCO [27] classes into four semantic classes. The *structure* class is used to construct the layout skeleton; the *geometrically inaccurate surfaces* are removed; the *objects* are used to fill holes, and the *stairs* are added back once the scene graph is created (Sec. 4.4). During prototype fitting (Sec. 4.3), we further distinguish between walls, ceilings, floors, and generic surfaces among the structures. Unlike mirrors, the outdoor classes are also used for window detection in Sec. 4.4 because they are typically visible through windows in indoor environments.

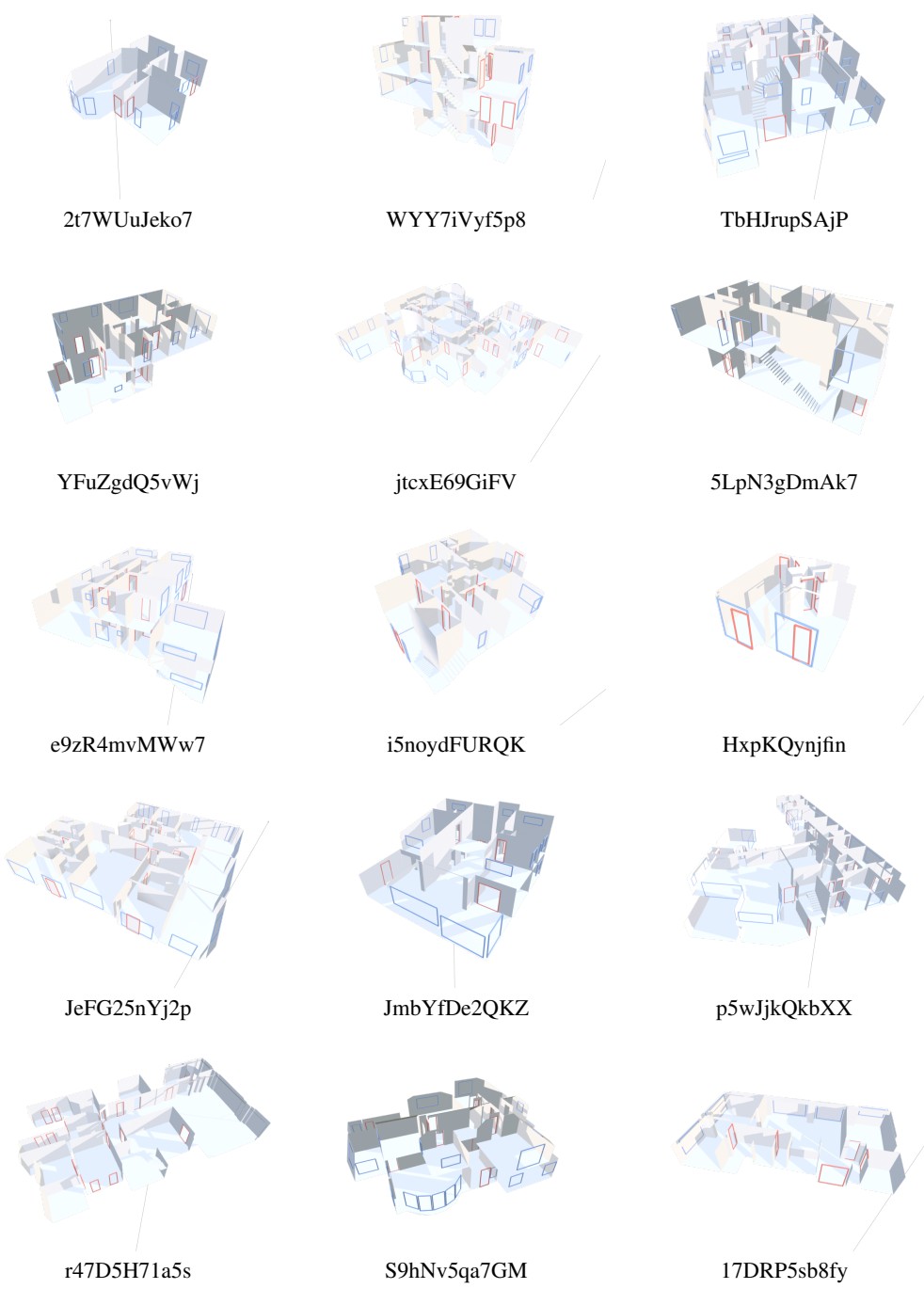

Figure 9: Visualizations and scene names of the scenes in the HOUSELAYOUT3D dataset. (The scene in Fig 1 was omitted).

## C   Implementation details for Layout Prototype Fitting

### C.1   Initialization of a Polygon Set from the Layout Skeleton

In Sec. 4.3 we fit a set of polygons to the layout skeleton to produce the *layout prototype*. Specifically, we sequentially fit one or more planes to each superpoint in the clustering using model fitting by random sample consensus (RANSAC [? ]). Then we extract polygons from the connected components of plane inliers (*i.e.* the points close to the planes): that is, we use the connectivity of the skeleton

**Require:** A mesh $M$ segmented into clusters $S = \{S_1, S_2, \ldots, S_n\}$
**Ensure:** A set of planar 3D polygons $\mathcal{P}$
 1: Mark every vertex in $M$ as `unassigned`
 2: Initialize $\mathcal{P} \leftarrow \emptyset$
 3: **while** there is an cluster with more than $K$ unassigned vertices **do**
 4:     Choose the cluster $S^*$ with the most unassigned vertices
 5:     Fit a plane to $S^*$ using RANSAC
 6:     Find all unassigned vertices in $M$ that lie close to the plane (the inliers)
 7:     Assign the vertices of the *connected component* of inliers that overlaps the most with $S^*$ to a new plane (connected w.r.t the mesh edges).
 8:     Extract polygon $P$ from the boundary of the triangles of the connected component, and add it to $\mathcal{P}$
 9: **end while**
10:
11: **return** $\mathcal{P}$

**Algorithm 1: Initialization of 3D Polygon Set.** We extract a set of planar 3D polygons from the clustered *layout skeleton* mesh by sequentially fitting planes to the superpoint clusters obtained in Sec. 4.2.

mesh to extract connected components of plane inliers. Then we take the boundary of each connected component as a polygon. Algorithm 1 describes the procedure.

## C.2 Implementation of a 3D Planar Polygon Set

In Sec. 4.3 we fit a set of 3D polygons to the vertices of the layout skeleton mesh using gradient descent.

For a set of $N$ polygons and $V$ vertices, we optimize the following parameters: 1) plane equations of shape $(N, 4)$ and 2) the vertex positions of shape $(V, 3)$. The parameters' gradients are computed using backpropagation.

**Implementation of 3D Planar Polygons as Triangle Meshes**   We build our implementation of a 3D polygon set on pytorch3d triangle meshes [36]. That is, we triangulate each polygon into triangular faces using Constrained Delaunay Triangulations [30] (CDT). To ensure that the polygons are planar, we for each polygon maintain a trainable plane equation. Upon accessing the (trainable) 3D position of a vertex, we first project the vertex to the plane constraint of the polygon it belongs to. Periodically, we update the original vertex position with the projected (constrained) position to avoid strong drift in the original vertex positions.

**Vertex Sharing**   We allow and encourage polygons to share vertices. If two vertices of different polygons are merged, this implies that we require the vertex to satisfy two plane equations. In that case, we project the vertex to the intersection line between the two polygons upon accessing its position.

Generally speaking, we store for each vertex up to three plane constraints based on the polygons it is part of, and project it to the closest point satisfying the constraints upon accessing its position.

To avoid training instabilities, we avoid merging vertices of near-parallel polygons.

**Re-Triangulation**   Upon adding triangle faces to polygons (projecting objects to the floor) or merging polygons, we re-triangulate the surface of each affected polygon using a CDT.

## D   Creation of a Scene Graph of 2D Floorplans: Detailed Description

In this step we use the prototype layout and its semantics to (1) identify the different levels (floors) of the building, (2) create a 2D layout (floorplan) of each level, and (3) segment each level into rooms, extracting a per-level 2D scene graph from each floor and (4) detect stairs to connect the individual levels. At a high level, the process can be summarized as follows:

**To identify building floors,** we use the floor-classified polygons of the layout prototype, merging close levels with similar heights.

**To create a 2D floorplan of each level,** we merge each level's floor polygon(s) with suitable ceiling polygons - since ceilings are rarely occluded by objects and thus are more robustly represented in the prototype layout.

**To segment each level into rooms** we apply Hov-SG [15]'s room segmentation algorithm on each 2D floorplan (and the walls of the prototype layout). The segmentation outputs a scene graph with rooms as nodes, and *openings* as edges. We consider an opening edge a *door* if its width is below 1.5m. Otherwise, we retain its edge but label it as *opening*. Furthermore, each room is associated with a room type (kitchen, office, ..).

**To identify stairs** we cluster connected components of the stair mesh extracted in Sec 4.2. For each component we add an edge to the scene graph between the rooms/floors it connects.

## D.1    Identifying Building Floors

We use the floor-classified polygons of the layout prototype and merge close levels with similar heights. Specifically, we create a graph where the nodes represent floor-classified polygons and add edges between polygons whose height differs by at most 50cm. Each connected component of the graph defines a floor level. For each level, we determine its average *elevation* from its assigned floor polygons.

## D.2    Creating a 2D Floorplan for each Level

We construct each level's 2D floorplan by computing the union of each level's floor polygon(s) with suitable ceiling polygons. Suitable ceiling polygons are identified by assigning each ceiling polygon to the closest next-lower floor polygon that is at least 1m below the ceiling's center. The level's *2D floorplan* is then constructed from the union of the ceilings and the floors of the level.

We further identify a level's *walls* by selecting wall-classified polygons that (1) intersect the floor's *2D floorplan* in BEV and (2) vertically intersect the height interval of $[0, 2.5]$m above the floor's *elevation*.

## D.3    Segmenting each Level into Rooms

We partition each level's *2D floorplan* into *rooms* using the level's *walls*. We do so by applying HovSG [15]'s morphology-based room segmentation algorithm twice: first with a bottleneck width of 2.5m and then with a bottleneck width of $1.5m$. The two-stage application has the benefit that cells (rooms) with a diameter between $1.5m$ and $2.5m$ can exist individually, yet larger cells with bottlenecks below 2.5m are separated.

Note that Hov-SG is a system designed for robotic navigation, and it does not reconstruct an explicit layout, neither in 2D nor in 3D. Originally, it obtains the 2D floorplan by simply thresholding the point density of the target floor.

We then follow HovSG in constructing a 2D *scene graph* with the *rooms* as nodes and the bottlenecks that split them as edges. We consider an edge a *door* if its bottleneck width is below 1.5m, and an *opening* otherwise. Each node has a *2D floorplan* consisting of a cell of the entire level's floorplan.

## D.4    Scene Graph Classification and Pruning

We follow HovSG in computing a single, CLIP-aligned feature per room. For this, we use OpenSeg [37] to compute pixel-aligned vision-language model features. Then we follow the same steps as in the mesh segmentation (Sec. 4.2) to project the features to our mesh vertices. The per-room feature is computed from the average mesh vertex features per room. We then use the CLIP embeddings to classify the rooms into *'bathroom'*, *'bedroom'*, *'living room'*, *'garage'*, *'entrance'*, *'kitchen'*, *'office'*, *'stairs'*, *'gym'*, *'classroom'*, *'spa/sauna'*, *'mirror'*, *'grass/bushes/trees'*, *'driveway'*, and *'veranda/terrace/balcony'*.

We then use this classification to remove leaf nodes of the scene graph belonging to one of the last five classes: This serves as an additional safeguard against the inclusion of outdoor spaces. (Additional to the segmentation performed in Sec. 4.2). Notably, this pruning step contributes to the performance improvement in Tab. 4 of the full method compared to the version without room segmentation.

### D.5 Stair Detection

We combine the *stair mesh* obtained in Sec. 4.2 with the floor segmentation from Sec. 4.4 to identify stairs and approximate them as simple 3D rectangles.

That is, we cluster the stair mesh (*i.e.*the sub-mesh of stair-classified vertices in Sec. 4.2) into connected components. Each component's vertices are now projected onto the horizontal plane and approximated by an oriented bounding rectangle $R$. We now assign the shorter edges of $R$ to rooms of the scene graph by (1) determining the heights of the shorter edge midpoints by interpolation on the 3D cluster vertices, (2) lifting the rectangle to 3D using the edge midpoint heights and (3) assigning it to the room with the shortest point-to-polygon distance ($D_{pp}$, defined in Sec. 4.3) to the edge midpoint. If this distance is greater than 50cm or both edges are assigned to the same room, we reject this cluster. Otherwise, we add a *stair* edge connecting the two rooms and store $R$ as its geometry.

### D.6 Handling Doors and Stairs during Floor Extrusion

The described room extrusion algorithm produces a closed shell for each room by extruding its 2D floorplan. To connect the rooms with doors and stairs, we introduce openings into these shells as follows:

To extrude doors, we approximate the room-splitting boundaries obtained by HovSG with oriented 2D bounding rectangles. From these bounding rectangles, we build a 3D doorframe of fixed height 2.10m consisting of four rectangular faces. During room extrusion, we ensure the doorframe remains empty by only adding wall triangles above the door for edge fragments inside the door's 2D bounding rectangle.

To extrude stairs, we first - in 2D - subtract the stair geometry from all rooms it intersects to avoid extruding overlapping regions. Then we extrude the stair geometry analogously to a room, with the difference that we as a last step adjust the height of the four floor corners in the resulting mesh to match the different levels it connects. We do not add walls for shared boundaries between stairs and rooms. For visualization, we add stair steps at a fixed stair step height on top of the otherwise pitched but flat floor of the stairs.

## E    Baseline Implementations

We compare our method to two recent, end-to-end trained baseline methods. Neither of the baselines is designed to predict multi-floor layouts. Adapting them to multiple floors is non-trivial. Hence, we use the ground-truth Matterport3D (MP3D) [7] segmentation of houses into levels and regions (loosely corresponding to rooms). We evaluate the baselines both on individual rooms and levels, concatenating the output.

### E.1    RoomFormer

RoomFormer [1] predicts a 2D layout based on point clouds. We hence create its input by sampling points from the surface of the mesh. The prediction is then lifted to 3D by assuming a planar floor/ceiling, whose height we determine based on the 5%-quantile of the distribution of the heights of the input points. Doors are assumed to extend from floor level to 2.10m above floor level. Windows are assumed to span 80% of the height of a wall (centered).

### E.2    SceneScript

SceneScript [2] predicts a set of 3D walls, doors, and windows based on *semidense* point clouds. We sample semi-dense from the input mesh by sampling points from the surface of the mesh where the norm of the surface gradient falls into the top 5%-quantile of all surface gradient norms. We further observe that the output improves when additionally sampling a small fraction of random points from all surfaces. We therefore incorporate random points into the input.

SceneScript neither predicts ceilings nor floors. We therefore infer one single floor and ceiling polygon respectively from the 2D Birds-Eye View convex hull of the output. To determine floor and ceiling height, we use the highest and lowest wall rectangles respectively.

