# OpenReview forum: "HouseLayout3D: A Benchmark and Training-free Baseline for 3D Layout Estimation in the Wild"
_NeurIPS.cc/2025/Datasets_and_Benchmarks_Track — NeurIPS 2025 Datasets and Benchmarks Track poster_

### Official Review · Reviewer_EG4s · 2025-07-02

**Rating:** 4
**Confidence:** 1

**Summary:**

This paper addresses the lack of datasets supporting multi-floor 3D layout estimation and provides a detailed explanation of how to construct a reliable multi-floor dataset to fill this gap. Compared to per-floor approaches, special attention is given to accurately modeling the connections between floors, particularly staircases. In addition, the method incorporates the learning of layout elements such as windows and doors, which contributes to generating high-quality 3D shapes.

**Dataset Code Accessibility:**

Yes

**Ethical Considerations:**

No, there are no or only very minor ethics concerns

**Final Justification:**

The authors have addressed my questions. As my concerns about this paper have been resolved, I give it a positive evaluation.

**Limitations Weaknesses:**

## Weaknesses including some questions

* Regarding large furniture such as closets: Were these included in the layout because of their large and rectangular shape? While that may be understandable, wouldn’t it be more appropriate to exclude such objects if the goal is to reconstruct the architectural layout of the building? I would appreciate it if you could clarify the rationale for including them.

* A question regarding the evaluation metrics: For structural components, rather than using general reconstruction metrics, would it be more suitable to treat rooms as the primary unit and use metrics such as IoU or Chamfer Distance? For example, RoomFormer explicitly reports IoU for layout evaluation—would a similar approach be applicable here?

* As far as I know, the Matterport3D dataset contains various types and shapes of staircases. Among them, spiral staircases and those with intermediate mid-floors are particularly unusual. How were such cases handled? Were these manually annotated or modeled?

* In addition, when filling holes in the floor, were you able to distinguish between regions that should remain empty (e.g., due to staircases) and regions that need to be filled (e.g., due to missing objects)? If so, how was this differentiation achieved?

## Minor typos

* “3d” should be capitalized as “3D”
* “w/a” appears in Figure 6 and should be revised for clarity

**Strengths Contributions:**

The introduction clearly identifies the limitations of existing datasets and presents a well-defined solution to address them. It is concise, well-structured, and easy to understand, making it an effective and engaging introduction to the paper.

---

> ### Author Rebuttal · Authors · 2025-07-30
>
> We thank the reviewer for their positive review. We appreciate that they recognize our proposed HouseLayout3D benchmark as a well-defined solution to address the need for a multi-floor 3D layout estimation dataset.
>
> Besides our main contribution, the HouseLayout3D dataset, our paper proposes a training-free baseline;
> below, we address the reviewers questions on the baseline.
>
> **Large Furniture** The reviewer asked to what extent large furniture is part of the layout.
> - In the manually labeled HouseLayout3D dataset, they are not included in the annotations. Annotators labeled the position of the underlying walls and floors as if these objects were not present, reflecting the intended architectural structure.
> - In our baseline method, MultiFloor3D, large objects such as carpets or curtains can be part of intermediate processing steps since the method operates solely on visible geometry. Removing them would create unwanted large holes in the Layout Skeleton (see Fig.3) that would require closing in subsequent steps (l.195).
>
> **Evaluation Metrics** The reviewer is curious about the choice of evaluation metrics. In this work, we adopt the metrics introduced in SceneScript, which are well suited for evaluating 3D layouts. The IoU metric, as suggested by the reviewer, is sometimes used in addition to F1 metrics in the context of 2D floorplan estimation. In general, compared to F1, IoU scores can be misleading as it may report high overlap even when the geometry is poorly aligned and the vertices are significantly offset. For example, a simple rectangle overlapping a complex geometric shape.
>
> **Staircases** The reviewer is curious how we handle different types of staircases as Matterport3D includes a variety of stair types and shapes. We annotate both rounded and straight stairs, by modeling the steps individually capturing the full geometry. If a staircase contains an intermediate platform, it is treated as two separate stair instances, one before and one after the platform. Evaluation is performed at the level of these connected stair instances.
>
> **Filling Holes** The reviewer asked how we differentiate between different types of holes in the ground plane. Our hole-filling strategy involves projecting objects onto the floor below. To distinguish between regions that should remain empty (e.g., stairwells) and regions that should be filled (e.g., gaps caused by missing objects), we rely on the $L_{empty}$ loss (Eq. 3). During optimization, if a projected polygon intersects with observed empty space - represented by line segments sampled between camera positions and backprojected depth image points - this intersection increases $L_{empty}$. This loss term is specifically designed to penalize occlusion of known empty space. As a result, polygons that overlap with such regions are penalized and gradually shrink during optimization, eventually being removed entirely.

---

> > ### Comment · Reviewer_EG4s · 2025-08-08
> >
> > I appreciate the authors’ efforts to address my concerns. Most of my questions have been resolved, and the dataset appears to offer significant value.
> >
> > That said, I would like to point out one remaining aspect. Due to certain limitations in the data generation methodology, it seems that large-scale objects had to be excluded. If this issue can be addressed in future work or through an updated version of the dataset, it would further enhance its value and impact.

---

### Official Review · Reviewer_nt4J · 2025-07-03

**Rating:** 4
**Confidence:** 3

**Summary:**

This paper proposes a benchmark dataset HouseLayout3D to provide large-scale, multi-floor real-world 3D building layouts for layout estimation, which fills the gap in current 3D layout benchmarks. In addition, the paper introduces a training-free method for 3D layout estimation, i.e. MultiFloor3D that leverages advances in 3D reconstruction and segmentation. Experiments on HouseLayout3D and ScanNet++ have shown the effectiveness of proposed method.

**Dataset Code Accessibility:**

Yes

**Ethical Considerations:**

No, there are no or only very minor ethics concerns

**Final Justification:**

Thanks to the authors for considering my review comments and providing detailed responses. I believe the authors' replies have basically addressed my concerns, and I lean towards accepting it, so I will maintain my original score.

**Limitations Weaknesses:**

1. Limited dataset scale and sources. Although the authors have mentioned that the size of proposed HouseLayout3D is comparable to ScanNet validation split, it is still relatively small compared with both common 3D scene datasets (i.e. ScanNet, ScanNet++) and previous 3D Layout Datasets (i.e. Structure3D and SceneCAD). Meanwhile, the dataset is built upon Matterport3D, which limits the diversity of data sources.
2. Unfair comparisons.
(1) The method MultiFloor3D consists of several other approaches or pre-trained models, which includes extra knowledge. Does this proposed MultiFloor3D still outperform other baselines without pre-trained models?
(2) The authors provide layout results only on the proposed dataset, and only depth results are provided on ScanNet++, which is not very convincing. Providing more layout results on other public datasets is welcomed.
3. The authors have mentioned the limitation that MultiFloor3D requires one or two hours, which restricts the application of this method.Besides, the memory cost is missing.
4. Minor: lack of failure case analysis.

**Strengths Contributions:**

1. This paper is well-written and easy to follow, with clear representation of both the proposed dataset and the method. Both tables and figures are clear.
2. The proposed benchmark is significant since it fills an important gap by providing large-scale multi-floor and real-world 3D layout annotations, which is valuable for further explorations on complex 3D layout estimation.
3. The proposed method for 3D layout estimation is simple but effective to perform over complex scenes, which do show improvement in HouseLayout3D against baselines.

---

> ### Author Rebuttal · Authors · 2025-07-30
>
> We thank the reviewer for the positive feedback and for recognizing our benchmark as **significant**, **filling an important gap**, and being **valuable for further explorations** on complex 3d layout estimation. Below we address the remaining concerns.
>
> **Dataset Diversity and Scale** The reviewer comments on the scale and diversity of the proposed dataset. While our real-world HouseLayout3D dataset is smaller than synthetically generated datasets (Structure3D) or small-scale (mostly) single-room datasets (ScanNet, SceneCAD) it is unique in the sense that it offers more diverse and challenging 3D layouts across complex large-scale spaces over multiple floors. In terms of diversity, our dataset is based on Matterport3D and therefore significantly more diverse than ScanNet which is limited to small-scale hotel rooms, offices and apartments (with mostly rectangular footprints). Beyond homes and offices, Matterport3D shows more challenging and varying non-cubical layouts and includes more exotic spaces such as churches, spas/saunas, gardens, balconies, stairs across multiple floors, large dining halls, swimming pools, museums etc.
>
> **Comparison to Baselines** The reviewer mentions that our training-free approach uses general knowledge from pre-trained models. This is correct. At the same time, both SceneScript and RoomFormer profit from highly specialized knowledge in the form of layout annotations. The key motivation behind our approach is to demonstrate that even a simple, training-free method can outperform such specialized models, particularly when applied to more challenging multi-floor datasets with complex layouts. Given recent developments in the field, it is increasingly reasonable to harness the extensive knowledge embedded in large pre-trained models, specifically in the era of foundation models. We expect to see more works following this direction.
>
> The reviewer also mentions that we provide results only on the proposed dataset. We also provide results on the ScanNet++ dataset (Table 3). However, due to the absence of ground-truth layout annotations in ScanNet++, we are limited to reporting depth-based metrics. That said, we agree with the reviewer that qualitative analysis would further strengthen our claims, and we will include qualitative results for ScanNet++ in the revised version of the paper.

---

> > ### Comment · Reviewer_nt4J · 2025-08-06
> >
> > Thanks to the authors for considering my review comments and providing detailed responses. I believe the authors' replies have basically addressed my concerns, and I lean towards accepting it, so I will maintain my original score.

---

### Official Review · Reviewer_G4L9 · 2025-07-03

**Rating:** 4
**Confidence:** 3

**Summary:**

This paper introduces HouseLayout3D, the first large-scale benchmark dataset specifically designed for 3D layout estimation in multi-floor buildings. It significantly expands the scope and complexity compared to previous datasets, which are typically limited to single-floor or synthetic scenes.
In addition to the dataset, the authors propose MultiFloor3D, a novel training-free baseline method for 3D layout estimation. xperiments on both the HouseLayout3D and ScanNet++ datasets demonstrate the superior generalization and robustness of MultiFloor3D.

**Dataset Code Accessibility:**

Yes

**Ethical Considerations:**

No, there are no or only very minor ethics concerns

**Limitations Weaknesses:**

1. The proposed MultiFloor3D pipeline requires 1–2 hours per scene to process, significantly slower than feed-forward methods like RoomFormer or SceneScript, which complete in a few minutes. This limits scalability and real-time applicability.
2. The pipeline depends on several external tools and pre-trained models (e.g., DN-Splatter, OneFormer, COLMAP), making it complex to implement and maintain. Errors or limitations in any stage may propagate to final layout predictions.
3. Although evaluated on ScanNet++, the generalization ability of the method across different domains (e.g., non-residential buildings, outdoor-indoor transitions, noisy sensors) is not extensively explored.
4. While the training-free nature is a strength in generalization, it also limits flexibility. The method relies on several handcrafted heuristics and fixed models (e.g., for segmentation), which may not adapt well to other domains or modalities (e.g., panoramas, LiDAR).
5. From a methodological novelty perspective, the work is more engineering-driven than algorithmically innovative. While effective in the current setting, the heavily engineered pipeline involves many stages (segmentation, mesh cleaning, fitting, graph conversion, extrusion, etc.), which may be unstable in the presence of domain shifts or partial data.

**Strengths Contributions:**

1. The paper presents HouseLayout3D, the first benchmark specifically targeting multi-floor, real-world 3D layout estimation. It fills a crucial gap in the literature where prior datasets focused on single-room or synthetic data. The dataset includes comprehensive CAD-style annotations for walls, floors, ceilings, doors (with opening directions), windows, and staircases, offering fine-grained supervision and enabling structured scene understanding.
2. The proposed method achieves state-of-the-art performance without any layout-specific training. This is particularly impressive given that other baselines rely on massive amounts of synthetic training data.

---

> ### Author Rebuttal · Authors · 2025-07-30
>
> We thank the reviewer for their constructive and detailed feedback. In particular, we appreciate the reviewer’s recognition that the proposed HouseLayout3D is the **first dataset** to **significantly expand the scope and complexity** compared to previous datasets typically limited to single-floor or synthetic scenes, **filling a crucial gap in the literature** towards multi-floor real-world 3D layout estimation.
>
> While our key contribution, the HouseLayout3D dataset, was very well received, the reviewer highlights a few shortcomings of the proposed training-free baseline. We address the main concerns below:
>
> - **Runtime** The reviewer notes that our training-free method has slower runtime compared to feed-forward method. This is correct, specialized feed-forward methods typically tend to offer superior inference speed compared to test-time optimization methods.
>
> - **Multi-stage pipeline** The reviewer highlights the potential drawbacks of our multi-stage pipeline, such as error propagation between stages. This is a valid concern. At the same time, a modular, interpretable pipeline offers the advantage of debuggability, each stage can be independently inspected and improved, which is often more challenging with end-to-end models.
>
> - **Generalization** The reviewer suggests that our method’s handcrafted heuristics may limit generalization to other modalities such as panoramas or LiDAR. We agree this is a limitation. However, it is worth noting that deep learning methods, particularly those operating on point clouds, can also struggle to generalize due to sensitivity to input distribution shifts (e.g., density, noise, sparsity). In contrast, our method may offer easier adaptability by tuning hyperparameters, without requiring new annotated datasets for retraining.
>
> In summary, while we acknowledge that all of the above are valid points, our goal was to demonstrate that a straightforward, training-free approach can already outperform specialized models in the context of more realistic, multi-floor environments. This result underscores both the challenges posed by our new benchmark dataset and the need for future research into methods capable of robust layout estimation under such complex conditions.

---

> > ### Comment · Reviewer_G4L9 · 2025-08-08
> >
> > Thanks for the response. After reading the rebuttal and other reviews, I have decided to maintain the same rating.

---

### Official Review · Reviewer_honL · 2025-07-21

**Rating:** 6
**Confidence:** 4

**Summary:**

This paper introduces HOUSELAYOUT3D, a new benchmark for 3D layout estimation focused on complex, multi-floor buildings from real-world scans. The authors identify that existing models, trained on simple synthetic data, fail to generalize. They propose MultiFloor3D, a training-free baseline method that leverages modern 3D reconstruction and segmentation techniques, and show that it outperforms current state-of-the-art methods.

**Dataset Code Accessibility:**

Yes

**Ethical Considerations:**

No, there are no or only very minor ethics concerns

**Limitations Weaknesses:**

1.  The method's multi-stage pipeline could be complex and prone to error propagation from earlier stages like 3D reconstruction.

2.  The approach can struggle to correctly handle outdoor elements visible through windows, leading to artifacts in the final layout.

**Strengths Contributions:**

1.  The work introduces the first benchmark dataset specifically for large-scale, multi-floor 3D layout estimation in real-world scenes.
2.  It proposes a novel and effective training-free method that significantly outperforms existing learning-based approaches.
3.  The public release of the high-quality annotated dataset and evaluation code is a valuable contribution to the research community.
4.  The paper clearly highlights the generalization failures of current state-of-the-art models, motivating new research directions.

---

> ### Author Rebuttal · Authors · 2025-07-30
>
> We thank the reviewer for the positive review and for recognizing the value of our **new benchmark dataset** for evaluating 3D layouts in complex, multi-floor buildings on real-world scans, including **high-quality annotations**.
>
> We also appreciate that the reviewer highlights the **novel and effective** training-free method **outperforming** specialized trained baselines, despite relying on a multi-stage pipeline. While this design can be sensitive to early-stage errors, it offers the advantage of making the individual stages easier to debug and improve compared to end-to-end models.
> Please also have a look at the response to reviewer *G4L9* on this aspect.
>
> Additionally, the reviewer highlights an **important phenomenon** revealed by our benchmark: current methods can produce partial 3D layouts of structures seen through windows, which can lead to unintended artifacts. This issue has not been observed in prior small-scale, single-room, or synthetic datasets, but is a challenge in real-world scenarios that our new real-world dataset exposes.

---

### Decision · Program_Chairs · 2025-09-18

**Decision:**

Accept (poster)

**Comment:**

The paper introduced HourseLayout3D, a real-world dataset for multi-floor indoor room layout. Building on top of this dataset, the paper further introduced a training-free method to estimate the 3D layout from input 2D images. Experiments demonstrated superior performance over learning-based methods, which failed to generalize beyond the training distribution.

The key strengths of this paper are: (1). real-world data sources with a focus on multi-floor settings; and (2). simple yet effective method for estimating 3D layout.

One potential weakness is that the dataset is relatively small compared to other datasets. After the rebuttal, all the reviewers recommend accepting the paper, the AC agrees with the reviewers, and recommends acceptance.